# Revisiting Dynamic Convolution via Matrix Decomposition

**Yunsheng Li[1], Yinpeng Chen[2], Xiyang Dai[2], Mengchen Liu[2], Dongdong Chen[2], Ye Yu[2],**
**Lu Yuan[2], Zicheng Liu[2], Mei Chen[2], Nuno Vasconcelos[1]**

[1] Department of Electrical and Computer Engineering, University of California San Diego
[2] Microsoft

`yul554@ucsd.edu, {yiche,xidai,mengcliu,dochen}@microsoft.com`
`{Yu.Ye,luyuan,zliu,Mei.Chen}@microsoft.com, nvasconcelos@ucsd.edu`

## Abstract

Recent research in dynamic convolution shows substantial performance boost for efficient CNNs, due to the adaptive aggregation of $K$ static convolution kernels. It has two limitations: (a) it increases the number of convolutional weights by $K$-times, and (b) the joint optimization of dynamic attention and static convolution kernels is challenging. In this paper, we revisit it from a new perspective of matrix decomposition and reveal the key issue is that dynamic convolution applies dynamic attention over channel groups after projecting into a higher dimensional latent space. To address this issue, we propose dynamic channel fusion to replace dynamic attention over channel groups. Dynamic channel fusion not only enables significant dimension reduction of the latent space, but also mitigates the joint optimization difficulty. As a result, our method is easier to train and requires significantly fewer parameters without sacrificing accuracy. Source code is at https://github.com/liyunsheng13/dcd.

## 1 Introduction

Dynamic convolution (Yang et al., 2019; Chen et al., 2020c) has recently become popular for the implementation of light-weight networks (Howard et al., 2017; Zhang et al., 2018b). Its ability to achieve significant performance gains with negligible computational cost has motivated its adoption for multiple vision tasks (Su et al., 2020; Chen et al., 2020b; Ma et al., 2020; Tian et al., 2020). The basic idea is to aggregate multiple convolution kernels dynamically, according to an input dependent attention mechanism, into a convolution weight matrix

$$\boldsymbol{W}(\boldsymbol{x}) = \sum_{k=1}^{K} \pi_k(\boldsymbol{x})\boldsymbol{W}_k \quad \text{s.t.} \quad 0 \leq \pi_k(\boldsymbol{x}) \leq 1, \sum_{k=1}^{K} \pi_k(\boldsymbol{x}) = 1, \tag{1}$$

where $K$ convolution kernels $\{\boldsymbol{W}_k\}$ are aggregated linearly with attention scores $\{\pi_k(\boldsymbol{x})\}$.

Dynamic convolution has two main limitations: (a) lack of compactness, due to the use of $K$ kernels, and (b) a challenging joint optimization of attention scores $\{\pi_k(\boldsymbol{x})\}$ and static kernels $\{\boldsymbol{W}_k\}$. Yang et al. (2019) proposed the use of a sigmoid layer to generate attention scores $\{\pi_k(\boldsymbol{x})\}$, leading to a significantly large space for the convolution kernel $\boldsymbol{W}(\boldsymbol{x})$ that makes the learning of attention scores $\{\pi_k(\boldsymbol{x})\}$ difficult. Chen et al. (2020c) replaced the sigmoid layer with a softmax function to compress the kernel space. However, small attention scores $\pi_k$ output by the softmax make the corresponding kernels $\boldsymbol{W}_k$ difficult to learn, especially in early training epochs, slowing training convergence. To mitigate these limitations, these two methods require additional constraints. For instance, Chen et al. (2020c) uses a large temperature in the softmax function to encourage near-uniform attention.

In this work, we revisit the two limitations via matrix decomposition. To expose the limitations, we reformulate dynamic convolution in terms of a set of residuals, re-defining the static kernels as

$$\boldsymbol{W}_k = \boldsymbol{W}_0 + \Delta\boldsymbol{W}_k, \quad k \in \{1, \dots, K\} \tag{2}$$

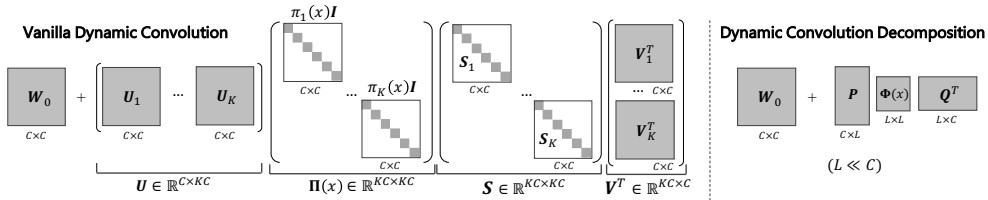

Figure 1: Dynamic convolution via matrix decomposition. **Left**: Reformulating the vanilla dynamic convolution by matrix decomposition (see Eq. 3). It applies ***dynamic attention*** $\mathbf{\Pi}(\boldsymbol{x})$ ***over channel groups*** in a ***high dimensional space*** ($\boldsymbol{SV}^T\boldsymbol{x} \in \mathbb{R}^{KC}$). **Right**: proposed dynamic convolution decomposition, which applies ***dynamic channel fusion*** $\mathbf{\Phi}(\boldsymbol{x})$ in a ***low dimensional space*** ($\boldsymbol{Q}^T\boldsymbol{x} \in \mathbb{R}^L$, $L \ll C$), resulting in a more compact model.

where $\boldsymbol{W}_0 = \frac{1}{K}\sum_{k=1}^K \boldsymbol{W}_k$ is the average kernel and $\Delta\boldsymbol{W}_k = \boldsymbol{W}_k - \boldsymbol{W}_0$ a residual weight matrix. Further decomposing the latter with an SVD, $\Delta\boldsymbol{W}_k = \boldsymbol{U}_k\boldsymbol{S}_k\boldsymbol{V}_k^T$, leads to

$$\boldsymbol{W}(\boldsymbol{x}) = \sum_{k=1}^K \pi_k(\boldsymbol{x})\boldsymbol{W}_0 + \sum_{k=1}^K \pi_k(\boldsymbol{x})\boldsymbol{U}_k\boldsymbol{S}_k\boldsymbol{V}_k^T = \boldsymbol{W}_0 + \boldsymbol{U}\mathbf{\Pi}(\boldsymbol{x})\boldsymbol{S}\boldsymbol{V}^T, \tag{3}$$

where $\boldsymbol{U} = [\boldsymbol{U}_1, \ldots, \boldsymbol{U}_K]$, $\boldsymbol{S} = diag(\boldsymbol{S}_1, \ldots, \boldsymbol{S}_K)$, $\boldsymbol{V} = [\boldsymbol{V}_1, \ldots, \boldsymbol{V}_K]$, and $\mathbf{\Pi}(\boldsymbol{x})$ stacks attention scores diagonally as $\mathbf{\Pi}(\boldsymbol{x}) = diag(\pi_1(\boldsymbol{x})\boldsymbol{I}, \ldots, \pi_K(\boldsymbol{x})\boldsymbol{I})$, where $\boldsymbol{I}$ is an identity matrix. This decomposition, illustrated in Figure 1, shows that the dynamic behavior of $\boldsymbol{W}(\boldsymbol{x})$ is implemented by the dynamic residual $\boldsymbol{U}\mathbf{\Pi}(\boldsymbol{x})\boldsymbol{S}\boldsymbol{V}^T$, which projects the input $\boldsymbol{x}$ to a higher dimensional space $\boldsymbol{SV}^T\boldsymbol{x}$ (from $C$ to $KC$ channels), applies dynamic attention $\mathbf{\Pi}(\boldsymbol{x})$ over channel groups, and reduces the dimension back to $C$ channels, through multiplication by $\boldsymbol{U}$. This suggests that the limitations of vanilla dynamic convolution are due to the use of *attention over channel groups,* which induces a high dimensional latent space, leading to small attention values that may suppress the learning of the corresponding channels.

To address this issue, we propose a *dynamic convolution decomposition* (DCD), that replaces dynamic attention over channel groups with *dynamic channel fusion*. The latter is based on a full dynamic matrix $\mathbf{\Phi}(\boldsymbol{x})$, of which each element $\phi_{i,j}(\boldsymbol{x})$ is a function of input $\boldsymbol{x}$. As shown in Figure 1-(right), the dynamic residual is implemented as the product $\boldsymbol{P}\mathbf{\Phi}(\boldsymbol{x})\boldsymbol{Q}^T$ of $\mathbf{\Phi}(\boldsymbol{x})$ and two static matrices $\boldsymbol{P}, \boldsymbol{Q}$, such that $\boldsymbol{Q}$ compresses the input into a low dimensional latent space, $\mathbf{\Phi}(\boldsymbol{x})$ dynamically fuses the channels in this space, and $\boldsymbol{P}$ expands the number of channels to the output space. The key innovation is that dynamic channel fusion with $\mathbf{\Phi}(\boldsymbol{x})$ enables a significant dimensionality reduction of the latent space ($\boldsymbol{Q}^T\boldsymbol{x} \in \mathbb{R}^L$, $L \ll C$). Hence the number of parameters in $\boldsymbol{P}, \boldsymbol{Q}$ is significantly reduced, when compared to $\boldsymbol{U}, \boldsymbol{V}$ of Eq. 3, resulting in a more compact model. Dynamic channel fusion also mitigates the joint optimization challenge of vanilla dynamic convolution, as each column of $\boldsymbol{P}, \boldsymbol{Q}$ is associated with multiple dynamic coefficients of $\mathbf{\Phi}(\boldsymbol{x})$. Hence, a few dynamic coefficients of small value are not sufficient to suppress the learning of static matrices $\boldsymbol{P}, \boldsymbol{Q}$. Experimental results show that DCD both significantly reduces the number of parameters and achieves higher accuracy than vanilla dynamic convolution, without requiring the additional constraints of (Yang et al., 2019; Chen et al., 2020c).

## 2   RELATED WORK

**Efficient CNNs:** MobileNet (Howard et al., 2017; Sandler et al., 2018; Howard et al., 2019) decomposes $k \times k$ convolution into a depthwise and a pointwise convolution. ShuffleNet (Zhang et al., 2018b; Ma et al., 2018) uses group convolution and channel shuffle to further simplify pointwise convolution. Further improvements of these architectures have been investigated recently. EfficientNet (Tan & Le, 2019a; Tan et al., 2020) finds a proper relationship between input resolution and width/depth of the network. Tan & Le (2019b) mix up multiple kernel sizes in a single convolution. Chen et al. (2020a) trades massive multiplications for much cheaper additions. Han et al. (2020) applies a series of cheap linear transformations to generate ghost feature maps. Zhou et al. (2020) flips the structure of inverted residual blocks to alleviate information loss. Yu et al. (2019) and Cai et al. (2019) train one network that supports multiple sub-networks of different complexities.

**Matrix Decomposition:** Lebedev et al. (2014) and Denton et al. (2014) use Canonical Polyadic decomposition (CPD) of convolution kernels to speed up networks, while Kim et al. (2015) investigates Tucker decompositions for the same purpose. More recently, Kossaifi et al. (2020) combines tensor decompositions with MobileNet to design efficient higher-order networks for video tasks, while Phan et al. (2020) proposes a stable CPD to deal with degeneracies of tensor decompositions during network training. Unlike DCD, which decomposes a convolutional kernel dynamically by adapting the core matrix to the input, these works all rely on static decompositions.

**Dynamic Neural Networks:** Dynamic networks boost representation power by adapting parameters or activation functions to the input. Ha et al. (2017) uses a secondary network to generate parameters for the main network. Hu et al. (2018) reweights channels by squeezing global context. Li et al. (2019) adapts attention over kernels of different sizes. Dynamic convolution (Yang et al., 2019; Chen et al., 2020c) aggregates multiple convolution kernels based on attention. Ma et al. (2020) uses grouped fully connected layer to generate convolutional weights directly. Chen et al. (2020b) extends dynamic convolution from spatial agnostic to spatial specific. Su et al. (2020) proposes dynamic group convolution that adaptively selects input channels to form groups. Tian et al. (2020) applies dynamic convolution to instance segmentation. Chen et al. (2020d) adapts slopes and intercepts of two linear functions in ReLU (Nair & Hinton, 2010; Jarrett et al., 2009).

## 3 DYNAMIC CONVOLUTION DECOMPOSITION

In this section, we introduce the *dynamic convolution decomposition* proposed to address the limitations of vanilla dynamic convolution. For conciseness, we assume a kernel $\boldsymbol{W}$ with the same number of input and output channels ($C_{in} = C_{out} = C$) and ignore bias terms. We focus on $1 \times 1$ convolution in this section and generalize the procedure to $k \times k$ convolution in the following section.

### 3.1 REVISITING VANILLA DYNAMIC CONVOLUTION

Vanilla dynamic convolution aggregates $K$ convolution kennels $\{\boldsymbol{W}_k\}$ with attention scores $\{\pi_k(\boldsymbol{x})\}$ (see Eq. 1). It can be reformulated as adding a dynamic residual to a static kernel, and the dynamic residual can be further decomposed by SVD (see Eq. 3), as shown in Figure 1. This has two limitations. First, the model is not compact. Essentially, *it expands the number of channels by a factor of $K$ and applies dynamic attention over $K$ channel groups*. The dynamic residual $\boldsymbol{U}\boldsymbol{\Pi}(\boldsymbol{x})\boldsymbol{S}\boldsymbol{V}^T$ is a $C \times C$ matrix, of maximum rank $C$, but sums $KC$ rank-1 matrices, since

$$\boldsymbol{W}(\boldsymbol{x}) = \boldsymbol{W}_0 + \boldsymbol{U}\boldsymbol{\Pi}(\boldsymbol{x})\boldsymbol{S}\boldsymbol{V}^T = \boldsymbol{W}_0 + \sum_{i=1}^{KC} \pi_{\lceil i/C \rceil}(\boldsymbol{x})\boldsymbol{u}_i s_{i,i}\boldsymbol{v}_i^T, \tag{4}$$

where $\boldsymbol{u}_i$ is the $i^{th}$ column vector of matrix $\boldsymbol{U}$, $\boldsymbol{v}_i$ is the $i^{th}$ column vector of matrix $\boldsymbol{V}$, $s_{i,i}$ is the $i^{th}$ diagonal entry of matrix $\boldsymbol{S}$ and $\lceil \cdot \rceil$ is ceiling operator. The static basis vectors $\boldsymbol{u}_i$ and $\boldsymbol{v}_i$ are not shared across different rank-1 matrices ($\pi_{\lceil i/C \rceil}(\boldsymbol{x})\boldsymbol{u}_i s_{i,i}\boldsymbol{v}_i^T$). This results in model redundancy. Second, it is difficult to jointly optimize static matrices $\boldsymbol{U}$, $\boldsymbol{V}$ and dynamic attention $\boldsymbol{\Pi}(\boldsymbol{x})$. This is because a small attention score $\pi_{\lceil i/C \rceil}$ may suppress the learning of corresponding columns $\boldsymbol{u}_i$, $\boldsymbol{v}_i$ in $\boldsymbol{U}$ and $\boldsymbol{V}$, especially in early training epochs (as shown in Chen et al. (2020c)).

### 3.2 DYNAMIC CHANNEL FUSION

We propose to address the limitations of the vanilla dynamic convolution with a dynamic channel fusion mechanism, implemented with a full matrix $\boldsymbol{\Phi}(\boldsymbol{x})$, where each element $\phi_{i,j}(\boldsymbol{x})$ is a function of input $\boldsymbol{x}$. $\boldsymbol{\Phi}(\boldsymbol{x})$ is a $L \times L$ matrix, dynamically fusing channels in the latent space $\mathbb{R}^L$. The key idea is to significantly reduce dimensionality in the latent space, $L \ll C$, to enable a more compact model. Dynamic convolution is implemented with dynamic channel fusion using

$$\boldsymbol{W}(\boldsymbol{x}) = \boldsymbol{W}_0 + \boldsymbol{P}\boldsymbol{\Phi}(\boldsymbol{x})\boldsymbol{Q}^T = \boldsymbol{W}_0 + \sum_{i=1}^{L}\sum_{j=1}^{L} \boldsymbol{p}_i \phi_{i,j}(\boldsymbol{x})\boldsymbol{q}_j^T, \tag{5}$$

where $\boldsymbol{Q} \in \mathbb{R}^{C \times L}$ compresses the input into a low dimensional space ($\boldsymbol{Q}^T\boldsymbol{x} \in \mathbb{R}^L$), the resulting $L$ channels are fused dynamically by $\boldsymbol{\Phi}(\boldsymbol{x}) \in \mathbb{R}^{L \times L}$ and expanded to the number of output channels

by $\boldsymbol{P} \in \mathbb{R}^{C \times L}$. This is denoted as **dynamic convolution decomposition** (DCD). The dimension $L$ of the latent space is constrained by $L^2 < C$. The default value of $L$ in this paper is empirically set to $\lfloor \frac{C}{2^{\lfloor \log_2 \sqrt{C} \rfloor}} \rfloor$, which means dividing $C$ by 2 repeatedly until it is less than $\sqrt{C}$.

With this new design, the number of static parameters is significantly reduced (i.e. $LC$ parameters in $\boldsymbol{P}$ or $\boldsymbol{Q}$ v.s. $KC^2$ parameters in $\boldsymbol{U}$ or $\boldsymbol{V}$, $L < \sqrt{C}$), resulting in a more compact model. Mathematically, the dynamic residual $\boldsymbol{P}\boldsymbol{\Phi}(\boldsymbol{x})\boldsymbol{Q}^T$ sums $L^2$ rank-1 matrices $\boldsymbol{p}_i \phi_{i,j}(\boldsymbol{x})\boldsymbol{q}_j^T$, where $\boldsymbol{p}_i$ is the $i^{th}$ column vector of $\boldsymbol{P}$, and $\boldsymbol{q}_j$ is the $j^{th}$ column vector of $\boldsymbol{Q}$. The constraint $L^2 < C$, guarantees that this number ($L^2$) is much smaller than the counterpart ($KC$) of vanilla dynamic convolution (see Eq. 4). Nevertheless, due to the use of a full matrix, dynamic channel fusion $\boldsymbol{\Phi}(\boldsymbol{x})$ retains the representation power needed to achieve good classification performance.

DCD also mitigates the joint optimization difficulty. Since each column of $\boldsymbol{P}$ (or $\boldsymbol{Q}$) is associated with multiple dynamic coefficients (e.g. $\boldsymbol{p}_i$ is related to $\phi_{i,1}, \ldots, \phi_{i,L}$), it is unlikely that the learning of $\boldsymbol{p}_i$ is suppressed by a few dynamic coefficients of small value.

In summary, DCD performs dynamic aggregation differently from vanilla dynamic convolution. Vanilla dynamic convolution uses a *shared dynamic attention* mechanism to aggregate *unshared static basis vectors* in a *high* dimensional latent space. In contrast, DCD uses an *unshared dynamic channel fusion* mechanism to aggregate *shared static basis vectors* in a *low* dimensional latent space.

### 3.3 MORE GENERAL FORMULATION

So far, we have focused on the dynamic residual and shown that dynamic channel fusion enables a compact implementation of dynamic convolution. We next discuss the static kernel $\boldsymbol{W}_0$. Originally, it is multiplied by a dynamic scalar $\sum_k \pi_k(\boldsymbol{x})$, which is canceled in Eq. 3 as attention scores sum to one. Relaxing the constraint $\sum_k \pi_k(\boldsymbol{x}) = 1$ results in the more general form

$$\boldsymbol{W}(\boldsymbol{x}) = \boldsymbol{\Lambda}(\boldsymbol{x})\boldsymbol{W}_0 + \boldsymbol{P}\boldsymbol{\Phi}(\boldsymbol{x})\boldsymbol{Q}^T, \tag{6}$$

where $\boldsymbol{\Lambda}(\boldsymbol{x})$ is a $C \times C$ diagonal matrix and $\lambda_{i,i}(\boldsymbol{x})$ a function of $\boldsymbol{x}$. In this way, $\boldsymbol{\Lambda}(\boldsymbol{x})$ implements channel-wise attention after the static kernel $\boldsymbol{W}_0$, generalizing Eq. 5 where $\boldsymbol{\Lambda}(\boldsymbol{x})$ is an identity matrix. Later, we will see that this generalization enables additional performance gains.

**Relation to Squeeze-and-Excitation (SE) (Hu et al., 2018):** The dynamic channel-wise attention mechanism implemented by $\boldsymbol{\Lambda}(\boldsymbol{x})$ is related to but *different* from SE. It is parallel to a convolution and shares the input with the convolution. It can be thought of as either a *dynamic convolution kernel* $\boldsymbol{y} = (\boldsymbol{\Lambda}(\boldsymbol{x})\boldsymbol{W}_0)\boldsymbol{x}$ or an input-dependent attention mechanism applied to the *output feature map* of the convolution $\boldsymbol{y} = \boldsymbol{\Lambda}(\boldsymbol{x})(\boldsymbol{W}_0\boldsymbol{x})$. Thus, its computational complexity is $\min(\mathcal{O}(C^2), \mathcal{O}(HWC))$, where $H$ and $W$ are height and width of the feature map.

In contrast, SE is placed *after* a convolution and uses the output of the convolution as input. It can only apply channel attention on the *output feature map* of the convolution as $\boldsymbol{y} = \boldsymbol{\Lambda}(\boldsymbol{z})\boldsymbol{z}$, where $\boldsymbol{z} = \boldsymbol{W}_0\boldsymbol{x}$. Its computational complexity is $\mathcal{O}(HWC)$. Clearly, SE requires more computation than dynamic channel-wise attention $\boldsymbol{\Lambda}(\boldsymbol{x})$ when the resolution of the feature map ($H \times W$) is high.

### 3.4 DYNAMIC CONVOLUTION DECOMPOSITION LAYER

**Implementation:** Figure 2 shows the diagram of a dynamic convolution decomposition (DCD) layer. It uses a light-weight dynamic branch to generate coefficients for both dynamic channel-wise attention $\boldsymbol{\Lambda}(\boldsymbol{x})$ and dynamic channel fusion $\boldsymbol{\Phi}(\boldsymbol{x})$. Similar to Squeeze-and-Excitation (Hu et al., 2018), the dynamic branch first applies average pooling to the input $\boldsymbol{x}$. This is followed by two fully connected (FC) layers with an activation layer between them. The first FC layer reduces the number of channels by $r$ and the second expands them into $C + L^2$ outputs ($C$ for $\boldsymbol{\Lambda}$ and $L^2$ for $\boldsymbol{\Phi}$). Eq. 6 is finally used to generate convolutional weights $\boldsymbol{W}(\boldsymbol{x})$. Similarly to a static convolution, a DCD layer also includes a batch normalization and an activation (e.g. ReLU) layer.

**Parameter Complexity:** DCD has similar FLOPs to the vanilla dynamic convolution. Here, we focus on parameter complexity. Static convolution and vanilla dynamic convolution require $C^2$ and $KC^2$ parameters, respectively. DCD requires $C^2$, $CL$, and $CL$ parameters for static matrices $\boldsymbol{W}_0$, $\boldsymbol{P}$ and $\boldsymbol{Q}$, respectively. An additional $(2C + L^2)\frac{C}{r}$ parameters are required by the dynamic branch

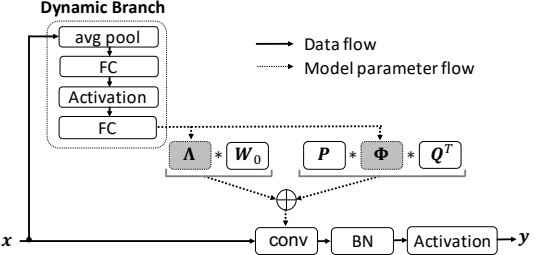

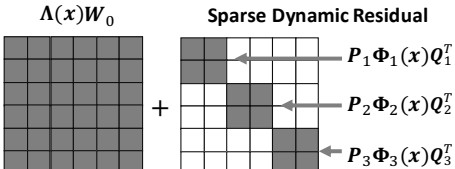

Figure 2: **Dynamic convolution decomposition layer**. The input $x$ first goes through a dynamic branch to generate $\mathbf{\Lambda}(x)$ and $\mathbf{\Phi}(x)$, and then to generate the convolution matrix $\boldsymbol{W}(x)$ using Eq. 6.

Figure 3: **Sparse dynamic residual**, which is represented as a diagonal block matrix. Each diagonal block is decomposed separately as $\boldsymbol{P}_b\mathbf{\Phi}_b\boldsymbol{Q}_b^T$. Note that the static kernel $\boldsymbol{W}_0$ is still a full size matrix.

to generate $\mathbf{\Lambda}(x)$ and $\mathbf{\Phi}(x)$, where $r$ is the reduction rate of the first FC layer. The total complexity is $C^2 + 2CL + (2C + L^2)\frac{C}{r}$. Since $L$ is constrained as $L^2 < C$, the complexity upper bound is $(1 + \frac{3}{r})C^2 + 2C\sqrt{C}$. When choosing $r = 16$, the complexity is about $1\frac{3}{16}C^2$. This is much less than what is typical for vanilla dynamic convolution ($4C^2$ in Chen et al. (2020c) and $8C^2$ in Yang et al. (2019)).

## 4 EXTENSIONS OF DYNAMIC CONVOLUTION DECOMPOSITION

In this section, we extend the dynamic decomposition of $1 \times 1$ convolution (Eq. 6) in three ways: (a) sparse dynamic residual where $\boldsymbol{P}\mathbf{\Phi}(x)\boldsymbol{Q}^T$ is a diagonal block matrix, (b) $k \times k$ depthwise convolution, and (c) $k \times k$ convolution. Here, $k$ refers to the kernel size.

### 4.1 DCD WITH SPARSE DYNAMIC RESIDUAL

The dynamic residual $\boldsymbol{P}\mathbf{\Phi}(x)\boldsymbol{Q}^T$ can be further simplified into a block-diagonal matrix of blocks $\boldsymbol{P}_b\mathbf{\Phi}_b(x)\boldsymbol{Q}_b^T, b \in \{1, \ldots, B\}$, leading to

$$\boldsymbol{W}(x) = \mathbf{\Lambda}(x)\boldsymbol{W}_0 + \bigoplus_{b=1}^{B} \boldsymbol{P}_b\mathbf{\Phi}_b(x)\boldsymbol{Q}_b^T, \tag{7}$$

where $\bigoplus_{i=1}^{n} A_i = diag(A_1, \ldots, A_n)$. This form has Eq. 6 as a special case, where $B = 1$. Note that the static kernel $\boldsymbol{W}_0$ is still a full matrix and only the dynamic residual is sparse (see Figure 3). We will show later that keeping as few as $\frac{1}{8}$ of the entries of the dynamic residual non-zero ($B = 8$) has a minimal performance degradation, still significantly outperforming a static kernel.

### 4.2 DCD OF $k \times k$ DEPTHWISE CONVOLUTION

The weights of a $k \times k$ depthwise convolution kernel form a $C \times k^2$ matrix. DCD can be generalized to such matrices by replacing in Eq. 6 the matrix $\boldsymbol{Q}$ (which squeezes the number of channels) with a matrix $\boldsymbol{R}$ (which squeezes the number of kernel elements)

$$\boldsymbol{W}(x) = \mathbf{\Lambda}(x)\boldsymbol{W}_0 + \boldsymbol{P}\mathbf{\Phi}(x)\boldsymbol{R}^T, \tag{8}$$

where $\boldsymbol{W}(x)$ and $\boldsymbol{W_0}$ are $C \times k^2$ matrices, $\mathbf{\Lambda}(x)$ is a diagonal $C \times C$ matrix that implements channel-wise attention, $\boldsymbol{R}$ is a $k^2 \times L_k$ matrix that reduces the number of kernel elements from $k^2$ to $L_k$, $\mathbf{\Phi}(x)$ is a $L_k \times L_k$ matrix that performs dynamic fusion along $L_k$ latent kernel elements and $\boldsymbol{P}$ is a $C \times L_k$ weight matrix for depthwise convolution over $L_k$ kernel elements. The default value of $L_k$ is $\lfloor k^2/2 \rfloor$. Since depthwise convolution is channel separable, $\mathbf{\Phi}(x)$ does not fuse channels, fusing instead $L_k$ latent kernel elements.

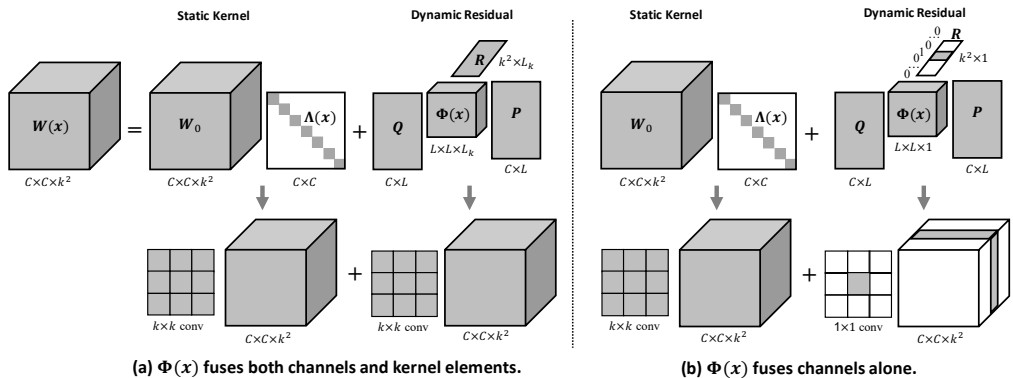

(a) $\Phi(x)$ fuses both channels and kernel elements.

(b) $\Phi(x)$ fuses channels alone.

Figure 4: The dynamic convolution decomposition for $k \times k$ convolution.

### 4.3 DCD of $k \times k$ Convolution

**Joint fusion of channels and kernel elements:** A $k \times k$ convolution kernel forms a $C \times C \times k^2$ tensor. DCD can be generalized to such tensors by extending Eq. 6 into a tensor form (see Figure 4)

$$W(x) = W_0 \times_2 \Lambda(x) + \Phi(x) \times_1 Q \times_2 P \times_3 R, \qquad (9)$$

where $\times_n$ refers to $n$-mode multiplication (Lathauwer et al., 2000), $W_0$ is a $C \times C \times k^2$ tensor, $\Lambda(x)$ is a diagonal $C \times C$ matrix that implements channel-wise attention, $Q$ is a $C \times L$ matrix that reduces the number of input channels from $C$ to $L$, $R$ is a $k^2 \times L_k$ matrix that reduces the number of kernel elements from $k^2$ to $L_k$, $\Phi(x)$ is a $L \times L \times L_k$ tensor that performs joint fusion of $L$ channels over $L_k$ latent kernel elements, and $P$ is a $C \times L$ matrix that expands the number of channels from $L$ to $C$. The numbers of latent channels $L$ and latent kernel elements $L_k$ are constrained by $L_k < k^2$ and $L^2 L_k \leq C$. Their default values are set empirically to $L_k = \lfloor k^2/2 \rfloor$, $L = \lfloor \frac{C/L_k}{2^{\lfloor log_2 \sqrt{C/L_k} \rfloor}} \rfloor$.

**Channel fusion alone:** We found that the fusion of channels $\Phi(x) \times_1 Q$ is more important than the fusion of kernel elements $\Phi(x) \times_3 R$. Therefore, we reduce $L_k$ to 1 and increase $L$ accordingly. $R$ is simplified into a one-hot vector $[0, \ldots, 0, 1, 0, \ldots, 0]^T$, where the '1' is located at the center (assuming that $k$ is an odd number). As illustrated in Figure 4-(b), the tensor of dynamic residual $\Phi(x) \times_1 Q \times_2 P \times_3 R$ only has one non-zero slice, which is equivalent to a $1 \times 1$ convolution. Therefore, the DCD of a $k \times k$ convolution is essentially adding a $1 \times 1$ dynamic residual to a static $k \times k$ kernel.

## 5 Experiments

In this section, we present the results of DCD on ImageNet classification (Deng et al., 2009). ImageNet has 1,000 classes with 1,281,167 training and 50,000 validation images. We also report ablation studies on different components of the approach.

All experiments are based on two network architectures: ResNet (He et al., 2016) and MobileNetV2 (Sandler et al., 2018). DCD is implemented on all convolutional layers of ResNet and all $1 \times 1$ convolutional layers of MobileNetV2. The reduction ratio $r$ is set to 16 for ResNet and MobileNetV2 $\times 1.0$, and to 8 for smaller models (MobileNetV2 $\times 0.5$ and $\times 0.35$). All models are trained by SGD with momentum 0.9. The batch size is 256 and remaining training parameters are as follows.

**ResNet:** The learning rate starts at 0.1 and is divided by 10 every 30 epochs. The model is trained with 100 epochs. Dropout (Srivastava et al., 2014) 0.1 is used only for ResNet-50.

**MobileNetV2:** The initial learning rate is 0.05 and decays to 0 in 300 epochs, according to a cosine function. Weight decay of 2e-5 and a dropout rate of 0.1 are also used. For MobileNetV2 $\times 1.0$, Mixup (Zhang et al., 2018a) and label smoothing are further added to avoid overfitting.

Table 1: **Different formulations** of dynamic convolution decomposition on ImageNet classification.

| Model | Params | MAdds | Top-1 |
|---|---|---|---|
| $W_0$ (static) | 2.0M | 97.0M | 65.4 |
| $\Lambda W_0$ | 2.4M | 97.4M | 68.2 |
| $W_0 + P\Phi Q^T$ | 2.7M | 104.4M | 69.2 |
| $\Lambda W_0 + P\Phi Q^T$ | 2.9M | 104.6M | **69.8** |

| Model | Params | MAdds | Top-1 |
|---|---|---|---|
| $W_0$ (static) | 11.1M | 1.81G | 70.4 |
| $\Lambda W_0$ | 11.7M | 1.81G | 71.5 |
| $W_0 + P\Phi Q^T$ | 13.6M | 1.83G | 72.8 |
| $\Lambda W_0 + P\Phi Q^T$ | 14.0M | 1.83G | **73.1** |

(a) **MobileNet V2** $\times 0.5$       (b) **ResNet-18**

## 5.1 INSPECTING DIFFERENT DCD FORMULATIONS

Table 1 summarizes the influence of different components (e.g. dynamic channel fusion $\Phi(x)$, dynamic channel-wise attention $\Lambda(x)$) of DCD on MobileNet V2 $\times 0.5$ and ResNet-18 performance. The table shows that both dynamic components, $\Lambda(x)$ and $\Phi(x)$ of Eq. 6. enhance accuracy substantially (+2.8% and +3.8% for MobileNetV2 $\times 0.5$, +1.1% and +2.4% for ResNet-18), when compared to the static baseline. Using dynamic channel fusion only ($W_0 + P\Phi Q^T$) has slightly more parameters, FLOPs, and accuracy than using dynamic channel-wise attention only ($\Lambda W_0$). The combination of the two mechanisms provides additional improvement.

## 5.2 ABLATIONS

A number of ablations were performed on MobileNet V2 $\times 0.5$ to analyze DCD performance in terms of two questions.

1. **How** does the dimension ($L$) of the latent space affect performance?

2. **How** do three DCD variants perform?

The default configuration is the general form of DCD (Eq. 6) with a full size dynamic residual ($B = 1$) for all pointwise convolution layers. The default latent space dimension is $L = \lfloor \frac{C}{2^{\lfloor log_2 \sqrt{C} \rfloor}} \rfloor$.

**Latent Space Dimension $L$:** The dynamic channel fusion matrix $\Phi(x)$ has size $L \times L$. Thus, $L$ controls both the representation and the parameter complexity of DCD. We adjust it by applying different multipliers to the default value of $L$. Table 2 shows the results of MobileNetV2 $\times 0.5$ for four multiplier values ranging from $\times 1.0$ to $\times 0.25$. As $L$ decreases, fewer parameters are required and the performance degrades slowly. Even with a very low dimensional latent space ($L \times 0.25$), DCD still outperforms the static baseline by 3.3% top-1 accuracy.

Table 2: **Dimension of the latent space** $L$ evaluated on ImageNet classification (MobileNetV2 $\times 0.5$ is used).

| Model | $L$ | Params | MAdds | Top-1 |
|---|---|---|---|---|
| static | - | 2.0M | 97.0M | 65.4 |
| DCD | $\times 0.25$ | 2.4M | 99.8M | 68.7 |
| | $\times 0.50$ | 2.5M | 101.3M | 69.0 |
| | $\times 0.75$ | 2.6M | 102.9M | 69.6 |
| | $\times 1.0$ | 2.9M | 104.6M | **69.8** |

**Number of Diagonal Blocks $B$ in the Dynamic Residual:** Table 3-(a) shows classification results for four values of $B$. The dynamic residual is a full matrix when $B = 1$, while only $\frac{1}{8}$ of its entries are non-zero for $B = 8$. Accuracy degrades slowly as the dynamic residual becomes sparser (increasing $B$). The largest performance drop happens when $B$ is changed from 1 to 2, as half of the weight matrix $W(x)$ becomes static. However, performance is still significantly better than that of the static baseline. The fact that even the sparsest $B = 8$ outperforms the static baseline by 2.9% (from 65.4% to 68.3%) demonstrates the representation power of the dynamic residual. In all cases, dynamic channel-wise attention $\Lambda(x)$ enables additional performance gains.

**DCD at Different Layers:** Table 3-(b) shows the results of implementing DCD for three different types of layers (a) DW: depthwise convolution (Eq. 8), (b) PW: pointwise convolution (Eq. 6), and (c) CLS: fully connected classifier, which is a special case of pointwise convolution (the input resolution is $1 \times 1$). Using DCD in any type of layer improves on the performance of the static baseline (+2.9% for depthwise convolution, +4.4% for pointwise convolution, and +1.2% for classifier). Combining DCD for both pointwise convolution and classifier achieves the best performance

Table 3: **Extensions of dynamic convolution decompostion (DCD)** evaluated on ImageNet classification (MobileNetV2 ×0.5 is used).

| Network | $B$ | Params | MAdds | Top-1 |
|---|---|---|---|---|
| $\boldsymbol{W}_0$ (static) | - | 2.0M | 97.0M | 65.4 |
| | 1 | 2.7M | 104.4M | **69.2** |
| $\boldsymbol{W}_0 + \boldsymbol{P}\boldsymbol{\Phi}\boldsymbol{Q}^T$ | 2 | 2.6M | 101.0M | 68.5 |
| | 4 | 2.5M | 99.1M | 68.4 |
| | 8 | 2.5M | 98.5M | 68.3 |
| | 1 | 2.9M | 104.6M | **69.8** |
| $\boldsymbol{\Lambda}\boldsymbol{W}_0 + \boldsymbol{P}\boldsymbol{\Phi}\boldsymbol{Q}^T$ | 2 | 2.8M | 101.3M | 68.9 |
| | 4 | 2.7M | 99.4M | 68.8 |
| | 8 | 2.7M | 98.8M | 68.5 |

| DW | PW | CLS | Params | MAdds | Top-1 |
|---|---|---|---|---|---|
| | | | 2.0M | 97.0M | 65.4 |
| ✓ | | | 2.4M | 97.5M | 68.3 |
| | ✓ | | 2.9M | 104.6M | 69.8 |
| | | ✓ | 2.2M | 97.2M | 66.6 |
| ✓ | | ✓ | 2.6M | 97.7M | 69.0 |
| ✓ | ✓ | | 3.3M | 105.1M | 69.6 |
| | ✓ | ✓ | 3.1M | 104.8M | **70.2** |
| ✓ | ✓ | ✓ | 3.5M | 105.3M | 70.0 |

(a) **Number of diagonal blocks** $B$ **in the dynamic residual.**

(b) **DCD at different layers**. DW, PW, and CLS indicate depthwise convolution, pointwise convolution and classifier respectively.

Table 4: Comparing DCD with the vanilla dynamic convolution CondConv (Yang et al., 2019) and DY-Conv (Chen et al., 2020c). ✶indicates the dynamic model with the fewest parameters (static model is not included). CondConv contains $K = 8$ kernels and DY-Conv contains $K = 4$ kernels.

| Width | Model | Params | MAdds | Top-1 |
|---|---|---|---|---|
| | static | 3.5M | 300.0M | 72.0 |
| ×1.0 | DY-Conv | 11.1M | 312.9M | **75.2** |
| | CondConv | 27.5M | 329.0M | 74.6 |
| | DCD (ours) | ✶5.5M | 326.0M | **75.2** |
| | static | 2.0M | 97.0M | 65.4 |
| ×0.5 | DY-Conv | 4.0M | 101.4M | 69.9 |
| | CondConv | 15.5M | 113.0M | 68.4 |
| | DCD (ours) | ✶3.1M | 104.8M | **70.2** |
| | static | 1.7M | 59.2M | 60.3 |
| ×0.35 | DY-Conv | 2.8M | 62.0M | 65.9 |
| | DCD (ours) | ✶2.3M | 63.1M | **66.6** |

| Depth | Model | Params | MAdds | Top-1 |
|---|---|---|---|---|
| ResNet-50 | static | 23.5M | 3.8G | 76.2 |
| | DCD (ours) | 30.7M | 3.9G | **77.9** |
| | static | 11.1M | 1.81G | 70.4 |
| ResNet-18 | DY-Conv | 42.7M | 1.85G | 72.7 |
| | DCD (ours) | ✶14.0M | 1.83G | **73.1** |
| | static | 5.2M | 0.89G | 63.5 |
| ResNet-10 | DY-Conv | 18.6M | 0.91G | 67.7 |
| | DCD (ours) | ✶6.5M | 0.90G | **68.8** |

(a) **MobileNetV2**.

(b) **ResNet**.

(+4.8%). We notice a performance drop (from 70.2% to 70.0%) when using DCD in all three types of layers. We believe this is due to overfitting, as it has higher training accuracy.

**Extension to** $3 \times 3$ **Convolution:** We use ResNet-18, which stacks 16 layers of $3 \times 3$ convolution, to study the $3 \times 3$ extension of DCD (see Section 4.3). Compared to the static baseline (70.4% top-1 accuracy), DCD with *joint fusion of channels and kernel elements* (Eq. 9) improves top-1 accuracy (71.3%) by 0.9%. The top-1 accuracy is further improved by 1.8% (73.1%), when using DCD with *channel fusion alone*, which transforms the dynamic residual as a $1 \times 1$ convolution matrix (see Figure 4-(b)). This demonstrates that dynamic fusion is more effective across channels than across kernel elements.

**Summary:** Based on the ablations above, DCD should be implemented with both dynamic channel fusion $\boldsymbol{\Phi}$ and dynamic channel-wise attention $\boldsymbol{\Lambda}$, the default latent space dimension $L$, and a full size residual $B = 1$. DCD is recommended for pointwise convolution and classifier layers in MobileNetV2. For $3 \times 3$ convolutions in ResNet, DCD should be implemented with channel fusion alone. The model can be made more compact, for a slight performance drop, by (a) removing dynamic channel-wise attention $\boldsymbol{\Lambda}$, (b) reducing the latent space dimension $L$, (c) using a sparser dynamic residual (increasing $B$), and (d) implementing DCD in depthwise convolution alone.

## 5.3 MAIN RESULTS

DCD was compared to the vanilla dynamic convolution (Yang et al., 2019; Chen et al., 2020c) for MobileNetV2 and ResNet, using the settings recommended above, with the results of

Table 4[1]. DCD significantly reduces the number of parameters while improving the performance of both network architectures. For MobileNetV2 ×1.0, DCD only requires 50% of the parameters of (Chen et al., 2020c) and 25% of the parameters of (Yang et al., 2019). For ResNet-18, it only requires 33% of the parameters of (Chen et al., 2020c), while achieving a 0.4% gain in top-1 accuracy. Although DCD requires slightly more MAdds than (Chen et al., 2020c), the increment is negligible. These results demonstate that DCD is more compact and effective.

Figure 5 compares DCD to DY-Conv (Chen et al., 2020c) in terms of training convergence. DY-Conv uses a large temperature in its softmax to alleviate the joint optimization difficulty and make training more efficient. Without any additional parameter tuning, DCD converges even faster than DY-Conv with a large temperature and achieves higher accuracy.

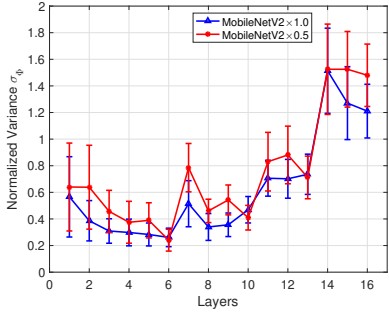

Figure 5: The comparison of training and validation error between DCD and DY-Conv on MobileNetV2 ×0.5. $\tau$ is the temperature in softmax. Best viewed in color.

### 5.4 ANALYSIS OF DYNAMIC CHANNEL FUSION

To validate the *dynamic* property, $\Phi(\boldsymbol{x})$ should have different values over different images. We measure this by averaging the variance of each entry $\sigma_\Phi = \sum_{i,j} \sigma_{i,j}/L^2$, where $\sigma_{i,j}$ is the variance of $\phi_{i,j}(\boldsymbol{x})$, over all validation images. To compare $\sigma_\Phi$ across layers, we normalize it by the variance of the corresponding input feature map. Figure 6 shows the normalized variance $\sigma_\Phi$ across layers in MobileNetV2. Clearly, the dynamic coefficients vary more in the higher layers. We believe this is because the higher layers encode more context information, providing more clues to adapt convolution weights.

Figure 6: Normalized variance of dynamic coefficients $\sigma_\Phi$ across layers in MobileNetV2 ×0.5 and ×1.0.

### 5.5 INFERENCE TIME

We use a single-threaded core AMD EPYC CPU 7551P (2.0 GHz) to measure running time (in milliseconds) on MobileNetV2 ×0.5 and ×1.0. Running time is calculated by averaging the inference time of 5,000 images with batch size 1. Both static baseline and DCD are implemented in PyTorch. Compared with the static baseline, DCD consumes about 8% more MAdds (97.0M vs 104.8M) and 14% more running time (91ms vs 104ms) for MobileNetV2 ×0.5. For MobileNetV2 ×1.0, DCD consumes 9% more MAdds (300.0M vs 326.0M) and 12% more running time (146ms vs 163ms). The overhead is higher in running time than MAdds. We believe this is because the optimizations of global average pooling and fully connected layers are not as efficient as convolution. This small penalty in inference time is justified by the DCD gains of 4.8% and 3.2% top-1 accuracy over MobileNetV2 ×0.5 and ×1.0 respectively.

## 6 CONCLUSION

In this paper, we have revisited dynamic convolution via matrix decomposition and demonstrated the limitations of dynamic attention over channel groups: it multiplies the number of parameters by $K$ and increases the difficulty of joint optimization. We proposed a dynamic convolution decomposition to address these issues. This applies dynamic channel fusion to significantly reduce the dimensionality of the latent space, resulting in a more compact model that is easier to learn with often improved accuracy. We hope that our work provides a deeper understanding of the gains recently observed for dynamic convolution.

---

[1]The baseline results are from the original papers. Our implementation, under the setup used for DCD, has either similar or slightly lower results, e.g. for MobileNetV2×1.0 the original paper reports 72.0%, while our implementation achieves 71.8%.

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
