# OpenReview forum: "Revisiting Dynamic Convolution via Matrix Decomposition"
_ICLR.cc/2021/Conference — ICLR 2021 Poster_

### Official Review · AnonReviewer2 · 2020-10-28
**Good paper, although not the easiest read**

**Rating:** 7
**Confidence:** 2

**Review:**

This is a little outside my area of expertise, but I found the paper interesting, proposing a different approach to dynamic convolutions based on matrix decomposition which requires fewer parameters to achieve similar accuracies as baselines while converging slightly quicker. The key ingredient is dynamic channel fusion which replaces softmax normalized attention as a mechanism used to combine information from different channels. The experiments look convincing and the results are somewhat useful, although additional work may be required to reduce the computational complexity (along with the number of parameters as shown in this work).

Some suggestions/questions:
- discuss the reasons behind vanilla dynamic convolution limitations earlier than in sec. 3.1
- if using dynamic conv in all layers leads to overfitting, why not try to tune regularization params to avoid this?

---

> ### Author Response · Authors · 2020-11-23
> **Author Response**
>
> Thanks for your thoughtful review. Below you will find our responses to your comments.
>
> **Q1: discuss the reasons behind vanilla dynamic convolution limitations earlier than in sec. 3.1**
>
> Thank reviewer 2 for this suggestion. We will discuss the reasons clearly in the introduction.
>
> **Q2: if using dynamic conv in all layers leads to overfitting, why not try to tune regularization params to avoid this?**
>
> We tried to tune the weight decay. Even though the gap between the training and validation accuracy shrinks when increasing the weight decay, the improvement in the validation accuracy is very minor.

---

### Official Review · AnonReviewer1 · 2020-10-28
**new perspective of understanding dynamic convolution**

**Rating:** 6
**Confidence:** 3

**Review:**

Authors presented a new perspective of matrix decompoistion to understand dynamic convolution and further proposed dynamic channel fusion which achieves both dimension reduction and good training convergence.

Extensive ablation study and experiments give good insight of the proposed method and helps understand its advantage. The experimental results on accuracy, training time, parameter counts, and flops demonstrate its superiority compared with other methods. Further analysis on the coefficients is interesting and also helps the understanding of the proposed method.

1. As for dynamic convolution decomposition (DCD), it is not clear how to address diversity of decomposition due to no additional constraint on P, phi, and Q. How does different kind of decomposition influnce the performance? If there is not any constraint on them, do they need special initialization?

2. In section 4.2, since k^2 is usually not very big, the saving on depth wise convolution is not very obvious.

3. Inference speed other than Flops should be reported to show its efficiency. Flops does not perfectly reflect its speed.

4. What about peak memory usage of the proposed method?

---

> ### Author Response · Authors · 2020-11-23
> **Author Response**
>
> Thanks for your thoughtful review. Below you will find our responses to your comments.
>
> **Q1: "... how to address diversity of decomposition due to no additional constraint on $\boldsymbol{P}$, $\boldsymbol{\Phi}$ and $\boldsymbol{Q}$. How does different kind of decomposition influence the performance? If there is not any constraint on them, do they need special initialization?..."**
>
> Although we do not apply extra constrains on $\boldsymbol{P}$, $\boldsymbol{\Phi}$, and $\boldsymbol{Q}$, the dynamic channel fusion matrix $\boldsymbol{\Phi}(\boldsymbol{x})$ enables diversity in the resulting convolution matrix $\boldsymbol{W}(\boldsymbol{x})$ as its entries are adapted to the input $\boldsymbol{x}$. Thank reviewer 1 for sharing this idea of adding extra diversity constraints in the decomposed matrices. We would like to try it in the future.
>
> Table 1a (in the submission draft) shows the performance for different decompositions. Compared to the static counterpart (65.4\% top-1 accuracy), dynamic channel-wise attention $\boldsymbol{W}(\boldsymbol{x})=\boldsymbol{\Lambda}(\boldsymbol{x})\boldsymbol{W}_0$ achieves 68.2\%, while dynamic channel fusion $\boldsymbol{W}(\boldsymbol{x})=\boldsymbol{W_0}+\boldsymbol{P}\boldsymbol{\Phi}(\boldsymbol{x})\boldsymbol{Q}^T$ achieves 69.2\%.
> The combination $\boldsymbol{W}(\boldsymbol{x})=\boldsymbol{\Lambda}(\boldsymbol{x})\boldsymbol{W_0}+\boldsymbol{P}\boldsymbol{\Phi}(\boldsymbol{x})\boldsymbol{Q}^T$ gains another 0.6\% to 69.8\%. In addition,
> when using dynamic channel fusion $\boldsymbol{P}\boldsymbol{\Phi}(\boldsymbol{x})\boldsymbol{Q}^T$, the performance is influenced by the dimension of the intermediate space $L$ ($\boldsymbol{\Phi}$ is of shape $L\times L$). Table 1b (in the submission draft) shows that the accuracy improves as $L$ increases.
>
> All parameters are initialized by Kaiming initialization, and the performance is stable over multiple runs.
>
> **Q2: "In section 4.2, since $k^2$ is usually not very big, the saving on depth wise convolution is not very obvious."**
>
> We agree that the parameter saving for depthwise convolution is not big. But it is important to show that DCD works for different types of convolutions: i.e. depthwise ($k \times k$), pointwise (1$\times$1), and standard ($k\times k$) convolutions. Interestingly, we found that using DCD on depthwise convolution **alone** achieves 2.9\% gain (from 65.4\% to 68.3\%) over the static counterpart with only 0.4M extra parameters and 0.5M extra MAdds (see Table 1d in the submission draft). It also outperforms using the vanilla dynamic convolution (DY-Conv) in depthwise convolution alone by 0.9\% (from 67.4\% to 68.3\%). Note that the result of using DY-Conv in depthwise convolution alone (67.4\%) is obtained from its original paper.
>
> **Q3: "Inference speed should be reported."**
>
> We use a single-threaded core of AMD EPYC CPU 7551P (2.0 GHz) to measure running time (in milliseconds). Here, MobileNetV2$\times0.5$ and MobileNetV2$\times1.0$ are used for testing. The running time is calculated by averaging the inference time of 5,000 images with batch size 1. Both static baseline and DCD are implemented in PyTorch.
>
> The results are shown in the two tables below. Compared with the static baseline, DCD consumes about 8\% more MAdds and 14\% more running time when using MobileNetV2$\times0.5$. When using MobileNetV2$\times1.0$, DCD consumes 9\% more MAdds and 12\% more running time. The overhead of running time is higher than MAdds. We believe this is because the optimizations of global average pooling and fully connected layers are not as efficient as convolution. Although a small amount of extra inference time is introduced, DCD achieves 4.8\% and 3.2\% improvement on top-1 accuracy for MobileNetV2$\times0.5$ and MobileNetV2$\times1.0$, respectively.
>
> |MobileNetV2$\times0.5$|MAdds|CPU (ms)|Top-1|
> |:-:|:-:|:-:|:-:|
> |static|97.0M|91|65.4|
> |DCD|104.8M|104|70.2|
>
> |MobileNetV2$\times1.0$|MAdds|CPU (ms)|Top-1|
> |:-:|:-:|:-:|:-:|
> |static|300.0M|146|72.0|
> |DCD|326.0M|163|75.2|
>
> **Q4: "What about peak memory usage of the proposed method?"**
>
> We test peak memory usage when training MobileNetV2$\times 0.5$ and MobileNetV2$\times 1.0$ with batch size 128 and image resolution $224 \times 224$ on a single GPU. The implementation is based on PyTorch and results are shown in the tables below. Compared to the static baseline, DCD consumes 17.9\% and 11.6\% more memory when using  MobileNetV2$\times 0.5$ and MobileNetV2$\times 1.0$, respectively.
>
> |MobileNetV2$\times 0.5$|Params|Peak Memory|Top-1|
> |:-:|:-:|:-:|:-:|
> | static | 2.0M | 5.6GB | 65.4 |
> | DCD | 3.1M | 6.6GB | 70.2 |
>
> |MobileNetV2$\times 1.0$|Params|Peak Memory|Top-1|
> |:-:|:-:|:-:|:-:|
> |static|3.4M|9.5GB|72.0|
> |DCD|5.5M|10.6GB|75.2|

---

### Official Review · AnonReviewer5 · 2020-11-02
**Insight into dynamic convolutions that reduces parameter count**

**Rating:** 6
**Confidence:** 3

**Review:**

This paper proposes a technique to reduce parameter count and improve training stability for dynamic convolutions using matrix decomposition. It looks at prior work (CondConv, DyConv) which aggregate multiple convolutional kernels via an attention score, and suggests this vanilla formulation is redundant since it sums $KC$ rank-1 matrices to create a rank-$C$ residual. The technique proposed in the paper (dynamic convolution decomposition) reduces the number of parameters via dimension reduction in the intermediate space, instead summing $L^2$ rank-1 matrices where $L^2 < C$. The authors provide experimental results on MobileNetV2 and ResNet comparing to prior works, as well as several ablations and possible extensions on their approach.

This paper helps address one key drawback of prior dynamic convolution approaches, which is that they require many more parameters than ordinary convolutions. However, this benefit needs to be better motivated by the paper. The goal of the prior dynamic convolution approaches was to create models that have higher performance and lower inference cost. In the case of MobileNetV2, while the parameter count is reduced, the inference cost is actually higher than the previous Dy-Conv approach to achieve the same performance. For this paper to be more compelling, it would be important to show reasons why reduced parameter count may be worth the increased inference cost, especially since from purely a parameter perspective, static scaling (with depth and width) seems to be more parameter-efficient. One possibility is that reducing the parameters can reduce actual latency on device at a given accuracy compared to prior dynamic convolution approaches due to requiring less memory bandwidth, but this is not shown or discussed in the paper.

The matrix decomposition lens suggested by this paper is interesting and gives insight into prior dynamic convolution approaches. How to generate kernels for dynamic convolutions is an important open question. To strengthen this section, it would be good to clarify the mathematical relationship between the proposed approach and the vanilla dynamic convolution approach in the CondConv and Dy-Conv papers. In particular, it would be good to give a comparison as to how the expressiveness of the proposed DCD approach compares to the expressiveness of the prior approaches. This would be important to give a complete picture of how the approach compares, beyond just parameter count. Also, the dynamic channel-wise attention seems to be related to Squeeze-and-Excitation, and it would be good to give an analysis for how these are related as well.

The experimental results of the paper can be improved by demonstrating the benefit of this approach on additional and more advanced mobile architectures, such as MobileNetV3. This would do a better job of illustrating the general benefit of the approach, especially in architectures that incorporate advances like Squeeze-and-Excitation. In a similar vein, comparing on a different task type (like object detection in CondConv or keypoint detection in Dy-Conv) could also help show more generality. The ablation experiments presented in this paper are relevant and help shed light on the approach and dynamic convolution.

Overall, the paper provides novel and relevant insight into dynamic convolutions, which leads to significantly reduced parameter count as shown in experiments. It could be strengthened by better motivating the importance of parameter reduction in this setting, better mathematical comparison to existing approaches, and more general experimental results.

---

> ### Author Response · Authors · 2020-11-23
> **Author Response (Part 2)**
>
> **Q3: "... the dynamic channel-wise attention seems to be related to Squeeze-and-Excitation..."**
>
> Thanks for this comment. The dynamic channel-wise attention is related to but **different** from Squeeze-and-Excitation (SE).
>
> In our method, the generation of channel-wise attention is **parallel** to a convolution and shares the input with the convolution. It can apply channel-wise attention either on the **convolution kernel** as $\boldsymbol{y}=(\boldsymbol{\Lambda}(\boldsymbol{x})\boldsymbol{W}_0)\boldsymbol{x}$ or on the **output feature map** of the convolution as $\boldsymbol{y}=\boldsymbol{\Lambda}(\boldsymbol{x})(\boldsymbol{W}_0\boldsymbol{x})$. The computational complexity of applying channel-wise attention is $\min(\mathcal{O}(C^2), \mathcal{O}(HWC))$, where $C$, $H$ and $W$ are the number of channels, height and width of the feature map.
>
> In contrast, SE is placed **after** a convolution and uses the output of the convolution as input.
> It can only apply channel-wise attention on the **output feature map** of the convolution as $\boldsymbol{y}=\boldsymbol{\Lambda}(\boldsymbol{z})\boldsymbol{z}$, where $\boldsymbol{z}=\boldsymbol{W}_0 \boldsymbol{x}$. The computational complexity of applying channel-wise attention is $\mathcal{O}(HWC)$. Clearly, SE requires more computation than DCD when the resolution of feature map ($H \times W$) is large.
>
> **Q4: "... experiments on more advanced mobile architectures, such as MobileNetV3... comparing on a different task type (like object detection in CondConv or keypoint detection in Dy-Conv) ..."**
>
> We conducted experiments on MobileNetV3-Small on ImageNet classification. The results are shown in the table below.
> The key conclusion holds: compared to DY-Conv, our DCD achieves similar accuracy with slightly more FLOPs (5\% more) but significantly fewer parameters (27\% fewer).
>
> |model|MAdds|Param|Top-1|
> |--|:-:|:-:|:-:|
> |static|66.0M|2.9M|67.4|
> |DY-Conv|68.5M|4.8M|70.3|
> |DCD|72.0M|3.5M|70.3|
>
> We also conducted experiments on COCO keypoint detection by using MobileNetV2 $\times 0.5$. We use the same training and testing setup as DY-Conv. The results are shown in the table below. The key conclusion holds: compared to DY-Conv, our DCD achieves similar AP with slightly more FLOPs (2\% more) but significantly fewer parameters (52\% fewer).
>
> |model|MAdds|Param|AP|
> |--|:-:|:-:|:-:|
> |static|794.8M|1.9M|59.2|
> |DY-Conv|807.4M|9.0M|62.8|
> |DCD|826.5M|4.3M|62.8|

---

> ### Author Response · Authors · 2020-11-23
> **Author Response (Part 1)**
>
> Thanks for your thoughtful review. Below you will find our responses to your comments.
>
> **Q1: "... this benefit (reducing parameters) needs to be better motivated ... it would be important to show reasons why reduced parameter count may be worth the increased inference cost... one possibility is that reducing the parameters can reduce actual latency on device at a given accuracy compared to prior dynamic convolution approaches due to requiring less memory bandwidth..."**
>
> We agree that the description of the motivation should be improved. We are motivated by the two limitations of the vanilla dynamic convolution: (a) **the number of parameters increases by $K$ times**, and (b) **the joint optimization between dynamic attention and static convolution kernels is challenging**. These two limitations are related, as the increasing number of parameters makes the joint optimization difficult. Therefore, reducing parameters not only results in a smaller model, but also mitigates the joint optimization difficulty in the vanilla dynamic convolution. Our goal is to design an elegant and easy-to-train solution, which can achieve a good **3-way balance between accuracy, FLOPs and the number of parameters**, instead of the 2-way balance only between accuracy and FLOPs in the vanilla dynamic convolution.
>
> Thank reviewer 5 for sharing the interesting idea of reducing actual latency on device due to fewer parameters. We would like to explore it in the future.
>
> **Q2: "...  clarify the mathematical relationship between the proposed approach and the vanilla dynamic convolution ... comparison of the expressiveness between DCD and the prior approaches..."**
>
> Both vanilla dynamic convolution and DCD have similar mathematical format: the resulting convolution matrix is the sum of a static convolution matrix $\boldsymbol{W_0}$ and a dynamic residual, which is decomposed into static matrices and one dynamic matrix, i.e. $\boldsymbol{W}(\boldsymbol{x})=\boldsymbol{W_0}+\boldsymbol{U}\boldsymbol{\Pi}(\boldsymbol{x})\boldsymbol{SV}^T$ in vanilla dynamic convolution *vs.* $\boldsymbol{W}(\boldsymbol{x})=\boldsymbol{W_0}+\boldsymbol{P}\boldsymbol{\Phi}(\boldsymbol{x})\boldsymbol{Q}^T$ in DCD. The dynamic residual ($\boldsymbol{U}\boldsymbol{\Pi}(\boldsymbol{x})\boldsymbol{SV}^T$ or $\boldsymbol{P}\boldsymbol{\Phi}(\boldsymbol{x})\boldsymbol{Q}^T$) can be considered as the dynamic aggregation of static basis vectors. Specifically, the vanilla dynamic convolution uses attention over channel groups $\boldsymbol{\Pi}(\boldsymbol{x})$ to aggregate basis vectors (or column vectors) in $\boldsymbol{U}$ and $\boldsymbol{V}$, while DCD uses dynamic channel fusion $\boldsymbol{\Phi}(\boldsymbol{x})$ to aggregate basis vectors (or column vectors) in $\boldsymbol{P}$ and $\boldsymbol{Q}$.
>
> However, DCD performs dynamic aggregation *differently* from the vanilla dynamic convolution. The vanilla dynamic convolution uses the **shared dynamic attention** to aggregate **unshared static basis vectors** in a **high** dimensional intermediate space. Specifically, each dynamic attention score  $\pi_k(\boldsymbol{x})$ is shared by a group of static basis vectors in $\boldsymbol{U}$ and $\boldsymbol{V}$, but each static basis vector is only related to an attention score. In contrast, our DCD solution uses the **unshared dynamic channel fusion** to aggregate **shared static basis vectors** in a **low** dimensional intermediate space. In particular, each static basis vector (in $\boldsymbol{P}$ and $\boldsymbol{Q}$) is shared by multiple dynamic entries $\phi_{ij}(\boldsymbol{x})$ in the dynamic channel fusion matrix $\boldsymbol{\Phi}(\boldsymbol{x})$, but each dynamic entry is only related to a basis vector. This key difference not only enables significant reduction in the number of parameters, but also mitigates the joint optimization challenge in the vanilla dynamic convolution.
>
> Thanks for this good suggestion. We will clarify this in the final draft.

---

### Official Review · AnonReviewer6 · 2020-11-03
**Paper Review**

**Rating:** 7
**Confidence:** 3

**Review:**

# Post-rebuttal updates
I've tentatively updated my review score from 6 to 7 for the following reasons:
* The authors clarified the relationship of their proposed method with Squeeze-and-Excite, and promised to add an explanation to their paper.
* The authors clarified the relationship between their method and CondConv in OpenReview comments, and promised to update the submission accordingly.
* The authors reported additional experiments on ResNet-18 and MobileNetV3(small). These new experiments help increase my confidence that the proposed method is broadly applicable.
* The authors provided inference time numbers for MobileNetV2 on CPU. AnonReviewer5 raised a very good point: that reducing the number of trainable parameters may lead to much faster models because of memory bandwidth considerations. I'm enthusiastic about this line of work, and think the inference time measurements provided in the rebuttal are a promising first step in this direction.

# Review Highlights
**Summary of Contributions:** The submission proposes a method for modifying a neural network so that the kernel for each convolution layer is computed dynamically based on its input. The authors claim -- and provide experiments to argue -- that this modification allows image classification networks to achieve better accuracy/size tradeoffs. The method is closely related to two existing pieces of work: CondConv (Yang 2019) and DY-Conv (Chen 2020). Compared with these existing methods, the submission's main contribution is a new function for dynamically generating/selecting convolutional kernels. The authors claim this new kernel-generating function achieves better accuracies with significantly fewer parameters than existing methods.

**Score Justification:** I've given this paper a borderline score. On the positive side: the submission does a good job of evaluating the proposed method, breaking down where improvements come from, and analyzing different variants of the method. Furthermore, the method does appear to provide better accuracy/size tradeoffs than earlier related work -- especially if we care about the number of model parameters. On the negative side: the submission appears to be an incremental improvement on existing work, and Row 2 (Model = $\Lambda W_0$) of Table 1(a) suggests that as much as 60%-70% of the accuracy gains of over a vanilla MobileNetV2 (0.5x) model can be attributed to Squeeze-and-Excite, which is a well-established technique.

**Method Details:** CondConv  and DY-Conv replace a fixed convolutional kernel $W$ with a weighted sum of kernels $\sum_{i=1}^{n}{\phi_i W_i}$. The kernels ${W_i}$ and the gating function which computes the scalars $\phi_1, \phi_2, \ldots, \phi_n$ are jointly learned via SGD. The submission instead replaces a rank-2 kernel $W$ for a fully connected layer with a function $\Lambda W_0 + P \Phi Q^T$. The fixed (input-independent) matrices $W_0, P, Q$ and the input-dependent gating functions which compute the matrix $\Phi$ and diagonal matrix $\Lambda$ are jointly learned via SGD. The authors present generalizations of their method to 1x1 convolutions, KxK convolutions, and depthwise convolutions. They also investigate the case where $\Phi$ is constrained to be sparse/block-diagonal.

**Note:** In the case where $\Phi$ is identically zero, I believe this formulation is equivalent or almost equivalent to Squeeze-and-Excite, where $\Lambda$ corresponds to the output of the Squeeze-and-Excite gating function.

**Pros:**
* Careful experimental evaluation / ablation experiments (Table 1). The submission evaluates several MobileNetV2 and ResNet models, both with and without the proposed modifications. It also evaluates several variants of the proposed method to quantify the relative importance of different design choices.

**Cons:**
* Some comparisons against related work could be improved/clarified. (See below.)
* Explanation of the proposed method can be improved. I initially had trouble understanding why SVD was used to motivate the proposed method, and initially also struggled to understand what the paper meant by "dynamic channel fusion". I found the direct formulation of the proposed method (Equation 5) to be much clearer and more explicit than the explanations that precede it.

**Reproducibility:** While the submission mentions many of the hyper-parameters used for model training, it would be helpful if the authors could include more details about how model weights were initialized at the start of training, since this wasn't obvious to me (and I couldn't find a discussion).

# Comparison against related work
While submission does a good job of highlighting related work (e.g,. Squeeze-and-Excite, Hyper-Networks), it primarily compares its method against two pieces of closely related work: CondConv and DY-Conv. I had the impression that comparisons against DY-Conv are executed much more carefully than comparisons against CondConv.

In particular:
* The submission's introduction claims that "the small attention score $\pi_k$ makes its corresponding kernel $W_k$ hard to learn" for both CondConv and DY-Conv. This makes sense to me for DY-Conv, since a softmax gating function is used to ensure that the attention weights $\alpha_i$ sum to 1. However, I don't understand why this should be the case for CondConv, which appears to use Sigmoid rather than Softmax gating.
* Table 2 compares results obtained using the submission's proposed parameterization against results reported by the CondConv and DY-Conv papers. However, it wasn't clear to me whether this comparison properly controlled for hyper-parameter tuning. For example: Table 2 reports that a static MobileNetV2 (1.0x) baseline model obtains an accuracy of 72.0%; this matches the baseline accuracy reported in the DyConv paper but is significantly higher than the baseline accuracy of 71.6% from the CondConv paper.

**Suggestions:** Comparisons against related work Table 1 should be updated to clearly show which of the models were trained with comparable hyper-parameters. In addition, the introduction should clarify which comparisons apply to DY-Conv and which comparisons apply to both DY-Conv and CondConv.

# Additional Notes

**Abstract:** *"[we] reveal the key issue is that dynamic convolution applies dynamic attention over channel groups after projecting into a higher dimensional intermediate space. To address this issue [...]"*
It wasn't obvious to me from reading this why applying dynamic convolution over attention groups was a bad thing. Why is this an "issue" or a problem that needs solving? This needs to be explained more clearly.

**Introduction:** The paper introduction claims that "joint optimization between attention scores and static kernels is challenging," but I didn't see any supporting evidence nearby. It would be helpful to add a citation for this claim, or a forward reference to supporting experiments.

**Section 3.4:** I tried to verify the theoretical analysis, and think it's correct. The implementation of implementation/parameter complexity looks right to me; parameter complexity analysis properly accounts for the cost of computing the activation/gating weights, which is good.

**Section 5.1:** Ablation experiments use MobileNetV2 (0.5x), which is a very small model. While it's great to have these experiments, I'm curious why the authors chose to focus on such a small model. How well do the findings hold up for a larger model such as MobileNetV2 (1.0x) or ResNet?

**Experiments:** Table 1 includes break-downs showing how the different components of the new DCD operator affect overall accuracy/FLOPS (including a static model baseline). Just weighting the kernel channels (analogous to squeeze-and-excite) accounts for a 2.8% accuracy gain; the dynamic components  of the convolution (the paper's new proposal) accounts for another 1.6% gain. This accounts for my earlier estimate that 60% - 70% of the gains came from Squeeze-and-Excite (since $2.8/(2.8 + 1.6) =  0.64$, or around 64%).

**Section 5.3:** Why is $\sigma_\phi$ normalized by the variance of the input feature map? (I had trouble understanding the paper's explanation; the explanation could be improved.)

---

> ### Author Response · Authors · 2020-11-23
> **Author Response (Part 3)**
>
> **Q3: More Details or Explanations**
>
> **"Abstract: ... why applying dynamic convolution over attention groups was a bad thing..."**
>
> Thanks for this comment. Our explanation could be improved. The problem is not applying dynamic attention over channel groups but **applying dynamic attention over channel groups after projecting into a higher dimensional intermediate space**. The channel expansion introduces unnecessary redundancy. This not only introduces more parameters, but also makes the joint optimization difficult.
>
> **"Ablation:... Why choose MobileNetV2 (0.5x)? ... How well do the findings hold up for a larger model such as MobileNetV2 (1.0$\times$) or ResNet"**
>
> We choose a smaller model for ablations by following prior works, as it is less time-consuming. CondConv uses MobileNetV1 0.25$\times$ and DY-Conv uses MobielNetV2 0.5$\times$ for ablations.
>
> We also conducted an ablation on ResNet-18 to study the dynamic channel fusion $\boldsymbol{\Phi}(\boldsymbol{x})$ and dynamic channel-wise attention $\boldsymbol{\Lambda}(\boldsymbol{x})$. The table below shows the results. The conclusion holds up. Due to the limited rebuttal time, we do not repeat all ablations on ResNet-18. But we will report it in the final version.
>
> |model|Top-1|
> |-|-:|
> |$\boldsymbol{W}_0$|70.4|
> |$\boldsymbol{\Lambda}(\boldsymbol{x})\boldsymbol{W}_0$|71.5|
> |$\boldsymbol{W}_0+\boldsymbol{P}\boldsymbol{\Phi}(\boldsymbol{x})\boldsymbol{Q}^T$|72.8|
> |$\boldsymbol{\Lambda}(\boldsymbol{x})\boldsymbol{W}_0+\boldsymbol{P}\boldsymbol{\Phi}(\boldsymbol{x})\boldsymbol{Q}^T$  |73.1|
>
> **"Section 5.3: Why is normalized by the variance of the input feature map?"**
>
> This is because the output variance of dynamic channel fusion $\boldsymbol{\Phi}(\boldsymbol{x})$ is related to two factors: (a) the variance of the input $\boldsymbol{x}$, and (b) the model parameters in $\boldsymbol{\Phi}(\cdot)$. Here, our goal is to compare the contribution of the model parameters in $\boldsymbol{\Phi}(\cdot)$ (the second factor) to the output variance across different layers. Therefore, we normalize the output variance by the input variance to eliminate the effect from the first factor. When performing analysis, we double check the input variance and observe it changes across different layers.
>
> **"Reproducibility: ... more details about how model weights were initialized ..."**
>
> Kaiming initialization is used for all layers.
>
> ------------------------------------------------------------------------------------------------------------------------------------------
>
> **Q4: Writing Related Questions**
>
> **"why SVD was used to motivate the proposed method?...struggled to understand what the paper meant by dynamic channel fusion ...  the direct formulation of the proposed method (Equation 5) is much clearer and more explicit ..."**
>
> Many thanks for these writing suggestions. We will integrate them in the final draft.
>
> **"Introduction: adding reference for the claim 'joint optimization between attention scores and static kernels is challenging'"**
>
> We will cite DY-Conv for this, as this claim is well explained in the beginning of section 4 of DY-Conv paper.

---

> ### Author Response · Authors · 2020-11-23
> **Author Response (Part 2)**
>
> **Q2: Comparison against related work**
>
> **"...comparisons against DY-Conv more carefully than CondConv... "**
>
> This is because DY-Conv is an improved version that addresses limitations in CondConv (see section 4.3 in DY-Conv). DY-Conv has less kernels per layer, smaller model size, less computation but achieves higher accuracy.
>
> **"...the small attention score $\pi_k$ makes its corresponding kernel $W_k$ hard to learn... why this should be the case for CondConv"**
>
> Thanks for this comment. We agree that the explanation should be improved. The small attention issue is related to DY-Conv but not to CondConv. CondConv and DY-Conv suffer from the joint optimization between attention and convolution kernels **from different perspectives**. In CondConv, using sigmoid results in a significantly larger space for the resulting convolution matrix, making the learning of **attention model** $\pi_k(\boldsymbol{x})$ difficult (this is discussed in Figure 4 and section 4.3 of DY-Conv paper). In DY-Conv, the small attention score $\pi_k$ makes its corresponding **convolution kernel** $\boldsymbol{W}_k$ hard to learn. In contrast, our method (DCD) effectively addresses this joint optimization problem.
>
> **"Table 2 compares results obtained using the submission's proposed parameterization against results reported by the CondConv and DY-Conv papers. However, it wasn't clear to me whether this comparison properly controlled for hyper-parameter tuning... Suggestions:...comparison of training setup (hyperparameters) with CondConv and DY-Conv..."**
>
> We compare the results of DCD (achieved by using the proposed training setup) to the results reported in the CondConv and DY-Conv papers, because different methods may have different optimal hyperparameters. A good comparison should allow each method to use its optimal setup. It is reasonable to assume that each work reports results by using its preferred setup. Below we list the training setups used in CondConv, DY-Conv and our DCD on MobileNetV2 as follows:
>
> **CondConv** uses the same training parameters as in the prior work ("Do Better ImageNet Models Transfer Better"). The network is trained with a batch size of 4096 using Nesterov momentum of 0.9 and weight decay 8e-5. It performs linear warmup to a learning rate of 1.6 over the first 10 epochs, and then continuously decayed the learning rate by a factor of 0.975 per epoch. It also uses dropout and data augmentation includes AutoAugment and Mixup.
>
> **DY-Conv** is trained with a batch size 256, using SGD optimizer with 0.9 momentum. The learning rate is 0.05 and decays to 0 in 300 epochs by a cosine function. The weight decay is 4e-5 and dropout rate is 0.2. Label smoothing and Mixup is used.
>
> **DCD** is trained with a batch size 256, using SGD optimizer with 0.9 momentum. The learning rate is 0.05 and decays to 0 in 300 epochs by a cosine function. The weight decay is 2e-5 and dropout rate is 0.1. We also use label smoothing and Mixup.
>
> Our DCD uses similar training setup to DY-Conv with small modification, as this setup requires less resource and simpler data augmentation (without using AutoAugment). It is *not* practical for us to use the training setup in CondConv.
>
> **"...baseline 72.0\% matches DY-Conv but is higher than 71.6\% in CondConv..."**
>
> We use the result reported in the original MobileNetV2 paper (72.0\%), even though our implementation only achieved 71.8\%.

---

> ### Author Response · Authors · 2020-11-23
> **Author Response (Part 1)**
>
> Thanks for your thoughtful review. Below you will find our responses to your comments.
>
> **Q1: Relation to SE "...appears to be an incremental improvement on existing work Squeeze-and-Excitation, that contributes 60\% to 70\% of the improvement"**
>
> The proposed dynamic channel-wise attention is related to but **different** from Squeeze-and-Excitation (SE).
>
> In our method, the generation of channel-wise attention is **parallel** to a convolution and shares the input with the convolution. It can apply channel-wise attention either on the **convolution kernel** as $\boldsymbol{y}=(\boldsymbol{\Lambda}(\boldsymbol{x})\boldsymbol{W}_0)\boldsymbol{x}$ or on the **output feature map** of the convolution as $\boldsymbol{y}=\boldsymbol{\Lambda}(\boldsymbol{x})(\boldsymbol{W}_0\boldsymbol{x})$. The computational complexity of applying channel-wise attention is $\min(\mathcal{O}(C^2), \mathcal{O}(HWC))$, where $C$, $H$, $W$ are the number of channels, height and width of the feature map.
>
> In contrast, SE is placed **after** a convolution and uses the output of the convolution as input.
> It can only apply channel-wise attention on the **output feature map** of the convolution as $\boldsymbol{y}=\boldsymbol{\Lambda}(\boldsymbol{z})\boldsymbol{z}$, where $\boldsymbol{z}=\boldsymbol{W}_0 \boldsymbol{x}$. The computational complexity of applying channel-wise attention is $\mathcal{O}(HWC)$. Clearly, SE requires more computation than DCD when the resolution of feature map ($H \times W$) is large.
>
> Next, we explain how different components contribute to the improvement from the static baseline (65.4\%) to the general form of DCD, i.e. $\boldsymbol{\Lambda}(\boldsymbol{x})\boldsymbol{W}_0+\boldsymbol{P}\boldsymbol{\Phi}(\boldsymbol{x})\boldsymbol{Q}^T$, which achieves 69.8\% top-1 accuracy. It is hard to claim that the channel-wise attention $\boldsymbol{\Lambda}(\boldsymbol{x})\boldsymbol{W}_0$ (top-1 accuracy 68.2\%) contributes $\frac{68.2-65.4}{69.8-65.4}$=63.6\%. This is because using the dynamic channel fusion alone $\boldsymbol{W}_0+\boldsymbol{P}\boldsymbol{\Phi}(\boldsymbol{x})\boldsymbol{Q}^T$ (without channel-wise attention $\boldsymbol{\Lambda}(\boldsymbol{x})$) achieves even higher accuracy 69.2\% (see Table 1 in the submission draft). If $\boldsymbol{\Lambda}(\boldsymbol{x})\boldsymbol{W}_0$ is considered to contribute 63.6\%, in the similar manner, the dynamic channel fusion $\boldsymbol{W}_0+\boldsymbol{P}\boldsymbol{\Phi}(\boldsymbol{x})\boldsymbol{Q}^T$ contributes $\frac{69.2-65.4}{69.8-65.4}$=86.4\% of the improvement and the channel-wise attention $\boldsymbol{\Lambda}(\boldsymbol{x})\boldsymbol{W}_0$ only contributes 13.6\% of the improvement. **Therefore, it is not accurate to claim the contribution as what is mentioned above, because the contribution of the two components are not fully complementary.**
> We believe the submission draft provides a fair statement: both parts significantly boost the accuracy compared to the static baseline. Using dynamic channel fusion alone has better performance than using dynamic channel-wise attention alone. The combination of these two provides additional improvement.

---

> > ### Comment · AnonReviewer6 · 2020-11-24
> > **Discussing relationship w/ Squeeze-and-Excite in main paper?**
> >
> > Thank you for your detailed comparison of your method against Squeeze-and-Excite. Since this is something that both AnonReviewer5 and I asked about: is the analysis from the first three paragraphs of your post something that you'd be willing to commit to adding to your paper if it's accepted to ICLR?
> >
> > Also: are you willing to commit to reporting the 71.8% accuracy from your MobileNetV2 reproduction in an updated version of your paper?

---

> > > ### Author Response · Authors · 2020-11-24
> > > **Re: Discussing relationship w/ Squeeze-and-Excite in main paper?**
> > >
> > > Yes, we will add the comparison between our dynamic channel-wise attention and Squeeze-and-Excitation (SE) in the final version.
> > >
> > > Regarding MobileNetV2 baseline, we will report (in the final version) 71.8\% with a footnote/remark indicating that this number is obtained from our own implementation while the result reported by the original paper was 72.0\%.

---

### Decision · Program_Chairs · 2021-01-07
**Final Decision**

**Decision:**

Accept (Poster)

**Comment:**

This paper improves the dynamic convolution operation by replacing the dynamic attention over channel groups with channel fusion in a low-dimensional space. It includes extensive experiments with reasonable baselines. Dynamic convolutions are a fruitful method for making convnets more efficient, and this paper further improves their efficiency and efficacy with a novel technique. Reviewers all agreed that the paper was clearly written (though some parts were improved after rebuttal).